# Synchronous Machine Winding Modeling Method Based on Broadband Characteristics

**Yu Chen** , **Xiaoqing Ji** and **Zhongyong Zhao** *

College of Engineering and Technology, Southwest University, Chongqing 400716, China;
cy1034429543@email.swu.edu.cn (Y.C.); memory0319@email.swu.edu.cn (X.J.)
* Correspondence: zhaozy1988@swu.edu.cn

**Abstract:** The accurate establishment of the equivalent circuit model of the synchronous machine windings' broadband characteristics is the basis for the study of high-frequency machine problems, such as winding fault diagnosis and electromagnetic interference prediction. Therefore, this paper proposes a modeling method for synchronous machine winding based on broadband characteristics. Firstly, the single-phase high-frequency lumped parameter circuit model of synchronous machine winding is introduced, then the broadband characteristics of the port are analyzed by using the state space model, and then the equivalent circuit parameters are identified by using an optimization algorithm combined with the measured broadband impedance characteristics of port. Finally, experimental verification and comparison experiments are carried out on a 5-kW synchronous machine. The experimental results show that the proposed modeling method identifies the impedance curve of the circuit parameters with a high degree of agreement with the measured impedance curve, which indicates that the modeling method is feasible. In addition, the comparative experimental results show that, compared with the engineering exploratory calculation method, the proposed parameter identification method has stronger adaptability to the measured data and a certain robustness. Compared with the black box model, the parameters of the proposed model have a certain physical meaning, and the agreement with the actual impedance characteristic curve is higher than that of the black box model.

**Keywords:** single-phase high-frequency lumped parameter model; optimization algorithm; state space; common mode and differential mode impedance

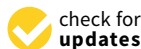



## 1. Introduction

A power system's safe operation determines whether human activities are carried out normally, so the reliability of a power system is a very relevant issue for people today [1–3]. At the heart of the power industry's energy supply are large-scale synchronous machines, of which regular operation determines whether a power system is adequately supplied with energy [4,5]. As a critical component of a synchronous machine, winding may cause defects, such as winding turn-to-turn faults and grounding short-circuits during synchronous machine operation [6]. With the continuous development of battery energy storage technology and increasingly severe environmental pollution problems, which have caused the constant increase in fuel vehicle emissions restrictions, it is bound to be the case that electric vehicles will replace traditional fuel vehicles [7,8]. The electromagnetic interference of a machine drive system is an essential factor affecting the operation of the machine [9]. Before starting research on the above-mentioned machine problems, modeling the equivalent broadband circuit of the winding is the fundamental base of this study.

At present, there are three broadly applicable methods for winding modeling: (1) Using the resonant unit of a black box model to simulate the information of each resonant point, and then cascade each resonant unit to construct a circuit model [9–11]; (2) a model is established using the physical properties of a machine, such as magnetic flux leakage, copper consumption, leakage current, etc. [12–14]; and (3) a whole machine winding's physical

structure is analyzed using finite element method, and then the winding is modeled using the analysis results [15].

For the first method, a fast-modeling method of the equivalent circuit of machine winding is proposed in References [9–11]. The method aims to measure the amplitude-frequency characteristics of the end impedance of machine winding, determine the resonance points on the amplitude-frequency characteristic curve, and each resonance point uses an *R*, *L*, *C* resonance unit as the equivalent circuit model. Finally, a topology structure diagram of a wide-band equivalent circuit of the machine is constructed. However, the machine winding model constructed using this method has no actual physical meaning, and the frequency band is low (1 Hz–1 MHz). In the second method [12,13], the physical meaning of the winding is firstly analyzed, and the fundamental circuit components that can represent the single-unit circuit are obtained. Then, the number of circuit units and the single-phase multi-unit circuit topology are obtained by analyzing the structure of the actual synchronous machine. Finally, the parameters of each fundamental circuit component are identified using the measured characteristics curve. However, the circuit parameters of the multi-stage unit constructed using this method are equal and cannot reflect the differences in winding in the winding process, and the interference of other phases cannot be prevented when measuring the broadband characteristic curve of single-phase winding; thus, the coincidence degree of its broadband equivalent characteristic curve is low. The literature [14] provides a method of modeling by indirectly measuring the characteristics of a port and uses the single-phase high-frequency lumped parameter model to build an equivalent model of the machine winding. Although this method uses common mode and differential mode measurement methods to prevent the interference of other phases to a certain extent when only measuring single-phase winding, this method uses an engineering exploratory calculation method to obtain parameters, which has inevitable human subjectivity and a low accuracy. In the third method [15], starting from the internal geometric structures of the synchronous machine, two-dimensional and three-dimensional solid modeling of the machine is carried out in actual scale in the electromagnetic finite element analysis software. Then, the actual physical parameters of the machine model are set, and the finite element analysis is carried out to obtain the electromagnetic parameters of the winding model, such as capacitance *C*, inductance *L*, and resistance *R*. Finally, by using these electromagnetic parameters and combining with the geometric distribution of the machine winding, a broadband equivalent circuit model of the winding can be drawn. Then the characteristic curve can be obtained. However, the mathematical derivation process of this method is complex, which needs to consider the detailed physical structure, size, material, and other parameters of the machine winding. This method not only increases the amount of work of researchers but also its accuracy cannot meet the higher requirements.

Therefore, it is necessary to propose a simple and fast machine winding model, which can prevent the interference of other phases in the measurement process, where the parameters have a certain practical physical significance to reflect the actual situation of the internal winding without losing a certain degree of accuracy so as to provide a research basis for the study of electromagnetic interference and other problems [9,16–18]. In conclusion, this paper proposes a modeling method of synchronous machine winding based on broadband characteristics, based on the existing research. This paper first introduces the structure of the single-phase high-frequency lumped parameter circuit model and the physical meaning of the corresponding parameters. The Kirchhoff's voltage laws(KVL) and Kirchhoff's current laws (KCL) equations of the model are used to construct a state space model to obtain the theoretical terminal characteristics. Then, the actual terminal characteristic curves of a synchronous machine are tested, and the circuit parameters are identified by using the proposed method. Finally, the performance of the proposed method is analyzed by comparing the experimental results with the identification results.

## 2. Theory Introduction

The flow chart of proposed method is shown in Figure 1.

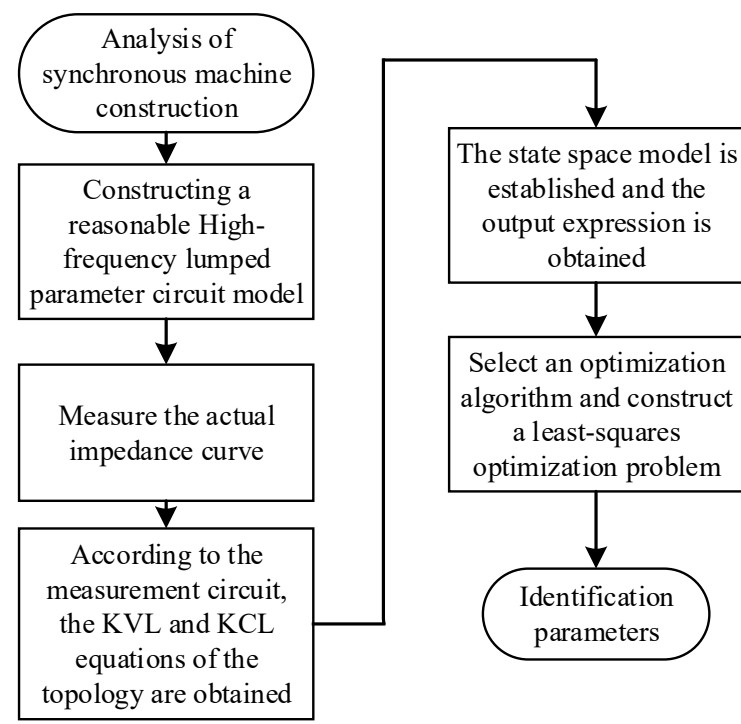

**Figure 1.** Flow chart of proposed method.

## 2.1. High-Frequency Lumped Parameter Circuit Model

According to References [16–18], a single-phase machine high-frequency lumped parameter circuit model derived based on common mode and differential mode impedance is constructed, and the equivalent model of single-phase winding is shown in Figure 2a. In the Figure 2a, according to Reference [17], $R_{g1}$ and $C_{g1}$ represent the parasitic resistance and capacitance between the stator winding and the motor frame; $R_{g2}$ and $C_{g2}$ represent the parasitic resistance and capacitance between the stator neutral and motor frame; and $R_e$, $R_{Cu}$, and $L_d$ represent the stator iron loss, stator winding copper loss, and stator winding leakage inductance, respectively. The parameters $R_1$, $L_1$, and $C_1$ are introduced to characterize the secondary resonance of the machine frequency characteristics, which is caused by the inter-turn stator winding capacitance and the skin effect. The circuit wiring diagrams for measuring the differential mode and common mode impedance of synchronous machines are shown in Figure 2b,c. Among them, $L_{zu}$ is the test lead inductance.

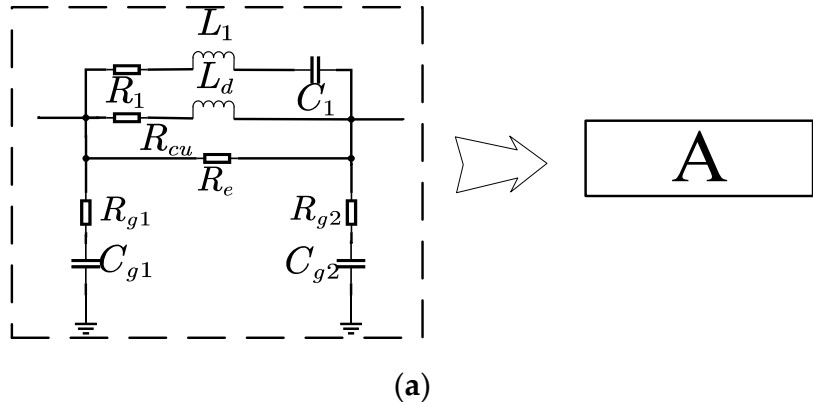

(**a**)

**Figure 2.** *Cont.*

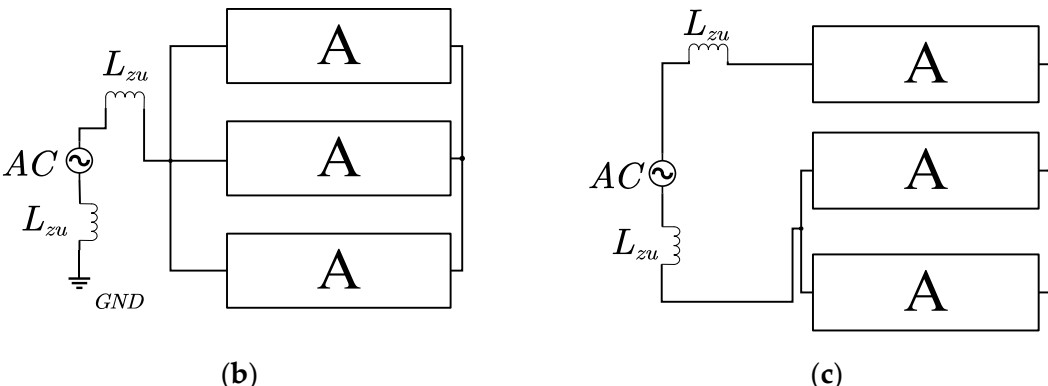

**Figure 2.** High-frequency lumped parameter circuit model of single-phase machine based on common mode and differential mode measurement:(**a**) single-phase winding model; (**b**)common mode impedance wiring diagram; and (**c**) differential mode impedance wiring diagram.

In order to obtain the value of each circuit parameter described above, the first step is to find the circuit equations that contain each parameter in the circuit topology shown in Figure 2b,c. When the above circuits are analyzed using the state space model, the capacitance voltage and inductance current in the single-phase lumped parameter circuit model are selected as the state variables. The KCL and KVL equations are written for the whole circuit, and these equations are finally integrated into matrices. The state space equation of the common mode impedance wiring diagram shown in Figure 2b is shown in Equation (1),

$$
\begin{pmatrix}
-1 & 0 & 0 & 0 & 0 & 0 \\
1 & 0 & -1 & 0 & 0 & -R_{Cu} \\
0 & -1 & 0 & 0 & R_{Cu} & 0 \\
0 & 0 & 0 & 0 & 1 & 0 \\
0 & 0 & 0 & \frac{1}{3} & -1 & -\left(1+\frac{R_{Cu}}{R_e}\right) \\
0 & 0 & 0 & 0 & 1 & 1+\frac{R_{Cu}}{R_e}
\end{pmatrix}
\begin{pmatrix}
V_1 \\ V_2 \\ V_3 \\ I_s \\ I_1 \\ I_2
\end{pmatrix}
+
\begin{pmatrix}
1 \\ 0 \\ 0 \\ 0 \\ 0 \\ 0
\end{pmatrix}
V_3 =
$$

$$
\begin{pmatrix}
R_{g1}C_{g1} & 0 & 0 & 2L_{zu} & 0 & 0 \\
-R_{g1}C_{g1} & 0 & C_{g2}R_{g2} & 0 & 0 & L_d \\
0 & R_1C_1 & 0 & 0 & L_1 & -L_d \\
0 & C_1 & 0 & 0 & 0 & 0 \\
C_{g1} & 0 & 0 & 0 & 0 & -\frac{L_d}{R_e} \\
0 & 0 & C_{g2} & 0 & 0 & -\frac{L_d}{R_e}
\end{pmatrix}
\begin{pmatrix}
\dot{V}_1 \\ \dot{V}_2 \\ \dot{V}_3 \\ \dot{I}_s \\ \dot{I}_1 \\ \dot{I}_2
\end{pmatrix}
\tag{1}
$$

where $V_s$, $V_1$, $V_2$, and $V_3$ are power supply voltage, the voltage on $C_{g1}$ of each phase, the voltage on $C_1$ of each phase and voltage on $C_{g2}$ of each phase respectively; $I_s$, $I_1$, and $I_2$ are current on $L_{zu}$, current on $L_1$ of each phase and current on $L_d$ of each phase, respectively.

The state space equation of the differential mode impedance wiring diagram shown in Figure 2c is shown in Equation (2),

$$\begin{pmatrix} -1 & 0 & 0 & 0 & 1 & 0 & 0 & 0 & 0 & 0 \\ 1 & -1 & 0 & 0 & 0 & 0 & -R_{Cu} & 0 & 0 & 0 \\ 0 & 0 & -1 & 0 & 0 & 0 & R_{Cu} & -R_1 & 0 & 0 \\ 0 & 0 & 0 & 0 & 0 & 0 & 0 & 1 & 0 & 0 \\ 0 & 0 & 0 & 0 & 0 & 0 & 0 & 0 & 0 & 1 \\ 0 & -1 & 0 & 0 & 1 & 0 & 0 & 0 & -R_{Cu} & 0 \\ 0 & 0 & 0 & 1 & 0 & 0 & 0 & 0 & -R_{Cu} & R_1 \\ 0 & 0 & 0 & 0 & 0 & 1 & -\frac{R_{Cu}+R_e}{R_e} & -1 & 0 & 0 \\ 0 & 0 & 0 & 0 & 0 & 0 & \frac{R_{Cu}+R_e}{R_e} & 1 & 2\frac{R_{Cu}+R_e}{R_e} & 2 \\ 0 & 0 & 0 & 0 & 0 & 1 & 0 & 0 & 2\frac{R_{Cu}+R_e}{R_e} & 2 \end{pmatrix} \begin{pmatrix} V_1 \\ V_2 \\ V_3 \\ V_4 \\ V_5 \\ I_s \\ I_1 \\ I_2 \\ I_3 \\ I_4 \end{pmatrix} + \begin{pmatrix} 1 \\ 0 \\ 0 \\ 0 \\ 0 \\ 0 \\ 0 \\ 0 \\ 0 \\ 0 \end{pmatrix} V_s =$$

$$\begin{pmatrix} C_{g1}R_{g1} & 0 & 0 & 0 & -C_{g1}R_{g1} & 2L_{zu} & 0 & 0 & 0 & 0 \\ -C_{g1}R_{g1} & C_{g2}R_{g2} & 0 & 0 & 0 & 0 & L_d & 0 & 0 & 0 \\ 0 & 0 & 0 & 0 & 0 & 0 & -L_d & L_1 & 0 & 0 \\ 0 & 0 & C_1 & 0 & 0 & 0 & 0 & 0 & 0 & 0 \\ 0 & 0 & 0 & C_1 & 0 & 0 & 0 & 0 & 0 & 0 \\ 0 & C_{g2}R_{g2} & 0 & 0 & -C_{g1}R_{g1} & 0 & 0 & 0 & L_d & 0 \\ 0 & 0 & 0 & 0 & 0 & 0 & 0 & 0 & L_d & -L_1 \\ C_{g1} & 0 & 0 & 0 & 0 & 0 & \frac{L_d}{R_e} & 0 & 0 & 0 \\ 0 & 3C_{g2} & 0 & 0 & 0 & 0 & -\frac{L_d}{R_e} & 0 & -2\frac{L_d}{R_e} & 0 \\ 0 & 0 & 0 & 0 & -2C_{g1} & 0 & 0 & -2\frac{L_d}{R_e} & 0 & 0 \end{pmatrix} \begin{pmatrix} \dot{V}_1 \\ \dot{V}_2 \\ \dot{V}_3 \\ \dot{V}_4 \\ \dot{V}_5 \\ \dot{I}_s \\ \dot{I}_1 \\ \dot{I}_2 \\ \dot{I}_3 \\ \dot{I}_4 \end{pmatrix} \tag{2}$$

where $V_s$, $V_1$, $V_2$, $V_3$, $V_4$, and $V_5$ are the power supply voltage, voltage on U-phase $C_{g1}$, voltage on each phase $C_{g2}$, voltage on U-phase $C_1$, voltage on V and W-phase $C_1$, and the voltage on V and W-phase $C_{g1}$, respectively; $I_s$, $I_1$, $I_2$, $I_3$, and $I_4$ are current on $L_{zu}$, current on U-phase $L_d$, current on U-phase $L_1$, current on V and W-phase $L_d$, and current on V and W-phase $L_1$, respectively.

Then the Equations of (1) and (2) are written as follows:

$$HI = C\dot{V} + GV \tag{3}$$

$$TV = RI + L\dot{I} \tag{4}$$

where $H$ is the current coefficient matrix, $T$ is the voltage coefficient matrix, $R$ is the resistance matrix, $L$ is the inductance matrix, $C$ is the capacitance matrix, and $G$ is the conductivity matrix. The input signal is $V_s$, and the measurement signal is $I_s$. $V_s$ needs to be extracted to calculate the transfer function.

$$T_0 V_s + T'V' = RI + L\dot{I} \tag{5}$$

$$HI = G_0 V_s + G_0 \dot{V}_s + G'V' + C'\dot{V'} \tag{6}$$

where $T_0$ is the first column matrix of $T$, $T'$ is the matrix after $T$ removes the first column, $V'$ is the matrix after $V$ removes $V_s$, $G_0$ is the first column of $G$, $C_0$ is the first column of $C$, $G$ removes $G_0$ to get $G'$, and $C$ removes $C_0$ to get $C'$.

The matrix is written as follows:

$$\begin{pmatrix} C' & 0 \\ 0 & L \end{pmatrix} \begin{pmatrix} \dot{V'} \\ \dot{I} \end{pmatrix} = \begin{pmatrix} -G'^{-1} & H \\ T' & -R \end{pmatrix} \begin{pmatrix} V' \\ I \end{pmatrix} + \begin{pmatrix} -G_0 \\ T_0 \end{pmatrix} V_s + \begin{pmatrix} C_0 \\ 0 \end{pmatrix} \dot{V}_s \tag{7}$$

$$A\dot{X} = BX + PV_s + Q\dot{V}_s \tag{8}$$

where $A = \begin{pmatrix} C' & 0 \\ 0 & L \end{pmatrix}, X = \begin{pmatrix} \dot{V}' \\ \dot{I} \end{pmatrix}, B = \begin{pmatrix} -G'^{-1} & H \\ T' & -R \end{pmatrix}, P = \begin{pmatrix} -G_0 \\ T_0 \end{pmatrix}, Q = \begin{pmatrix} C_0 \\ 0 \end{pmatrix}.$

$$I_s = MX \tag{9}$$

Laplace transformations are used for Equations (8) and (9), and the transfer function can be obtained by eliminating state variable $X$:

$$Z(s) = \frac{V_s}{I_s} = \frac{1}{M(sA - B)^{-1}(P + sQ)} \tag{10}$$

### 2.2. Parameter Identification Based on Optimization Algorithm

When identifying the model's parameters, the theoretical model with the experimental data are usually combined to identify the parameter value [19–26]. As an essential method of parameter identification, the optimization algorithm has a strong search performance for a specific parameter range, and it has high adaptability for practical problems [19,21,23,24]. In addition, it can visualize the performance of identification results through the value of the fitness function. Therefore, this paper chooses an optimization algorithm as a tool for parameter identification.

First of all, the particle swarm optimization (PSO) algorithm is selected as the identification algorithm. The circuit constructed in this paper is relatively simple, and it does not need an algorithm with a strong search ability and complicated calculations. Second, because the PSO algorithm has few adjustable hyperparameters, the algorithm structure is relatively simple compared to other intelligent algorithms (simulated annealing algorithm, genetic algorithm, immune algorithm, etc.). It is also easy to implement in engineering and has a faster convergence speed [20,22,23]. Third, the algorithm structure of PSO is simple and it has high universality. Fourth, there are a large number of identification parameters in this study. The number of solutions of the problem can be directly taken as the dimension of the particle in the PSO algorithm. Thus, compared with other algorithms, the PSO algorithm is less affected by the dimension of the problem [20,22,23].

In the particle swarm optimization algorithm, each particle is the random solution of the target to be optimized, and the objective function to be optimized determines the individual fitness of each particle. Each particle has its own speed and direction in the search process, and the particle will follow the current optimal particle to search in the solution space. The search method of the particles in the solution space is shown in Figure 3, $v$ is the search speed of particles in the solution space, $x$ is the example position, $p_{best}$ is the best position in history, $g_{best}$ is the best position at present, and $d$ is the number of iterations. A flow chart of the particle swarm optimization algorithm is shown in Figure 4.

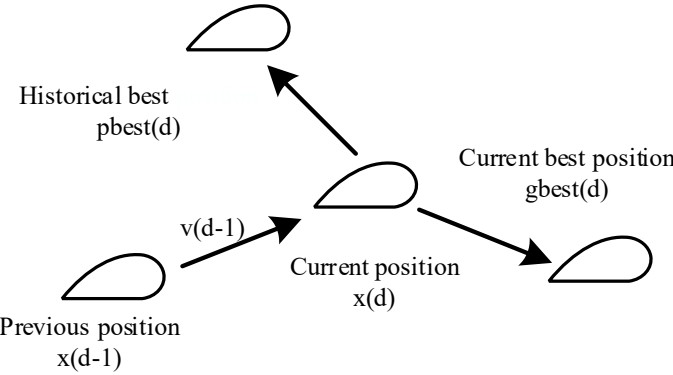

**Figure 3.** Solution space of particle swarm optimization.

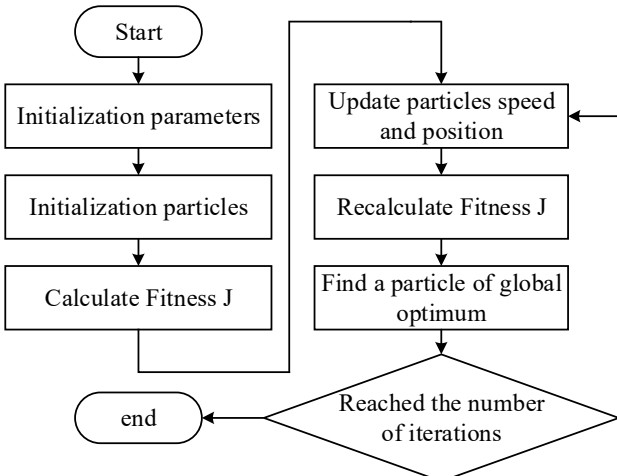

**Figure 4.** Flow chart of particle swarm optimization.

The impedance is sampled when the frequency is equal to the frequency of each point of the actual measurement curve. Then a least-square optimization problem is formed between the impedance and the actual measurement value. The least-square optimization problem is taken as a fitness function. The smaller the value of the fitness function is, the closer the simulation curve is to the actual value. For the constructed model, a fitness function is constructed between the result of Equation (10) and the measurement curve:

$$J = \sum_{f=1}^{N} \left( \frac{Z_{cm}(f) - Z_{cm1}(f)}{Z_{cm}(f)} \right)^2 + \sum_{f=1}^{N} \left( \frac{Z_{dm}(f) - Z_{dm1}(f)}{Z_{dm}(f)} \right)^2 \tag{11}$$

where $f$ is the frequency, $Z_{cm}$, $Z_{cm1}$, $Z_{dm}$, and $Z_{dm1}$ are the common mode impedance measurement value and the identification value, and the differential mode impedance measurement value and the identification value, respectively.

## 3. Experimental Verification

The impedance characteristic curve of a 5-kW, three-phase, salient pole synchronous generator machine winding was measured by using a KEYSIGHT impedance analyzer (model E4990A) and the sweep frequency range was 20 Hz–120 MHz. The connection diagram is shown in Figure 5. The common mode and the differential mode measurements can prevent the influence of other phases in a single-phase measurement, and it can contain more characteristic curve information, which is closer to the actual machine working characteristic curve than single-phase measurements [9,14,16–18].

Because there are too many parameters to search, to avoid falling into the local optimal solution, some exploratory calculations need to be carried out before the parameter identification, this limits the upper and lower limits of the parameters to be identified to a certain range, and then the parameter search is carried out, thus the probability of the optimal value will be significantly improved. In order to highlight the superiority and adaptability of the proposed method, this paper only limits the lower limit of all parameters to be greater than 0 and then brings the result of the Equation (10) into the Equation (11) to construct the fitness function.

The change in the fitness function is shown in Figure 6. Figure 6 shows the fitness Equation (11) trend between the impedance characteristic curve formed by the optimal solution, corresponding to each iteration, and the actual measured value during the iteration process. The smaller the fitness function value is, the closer the impedance characteristic curve simulated by the optimal solution is to the measured curve [19]. According to Figure 6, it can be seen that in the initial stage of a PSO search, the fitness of PSO is sharply reduced, which indicates that the impedance value calculated by the parameters searched

by the algorithm at each frequency point is very close to the value of the measured curve at each frequency point. However, in the later stage of the algorithm, there is no change in fitness, which indicates that the local change in the parameters identified in the later stage of the algorithm under the previous search results cannot significantly affect the change in the fitness value. This phenomenon also indicates that the search results of the algorithm are close to the optimal solution. To a certain extent, it shows that PSO is indeed suitable for multi-parameters identification problems and can achieve an optimal solution close to that of the actual measurement curve in a small number of iterations.

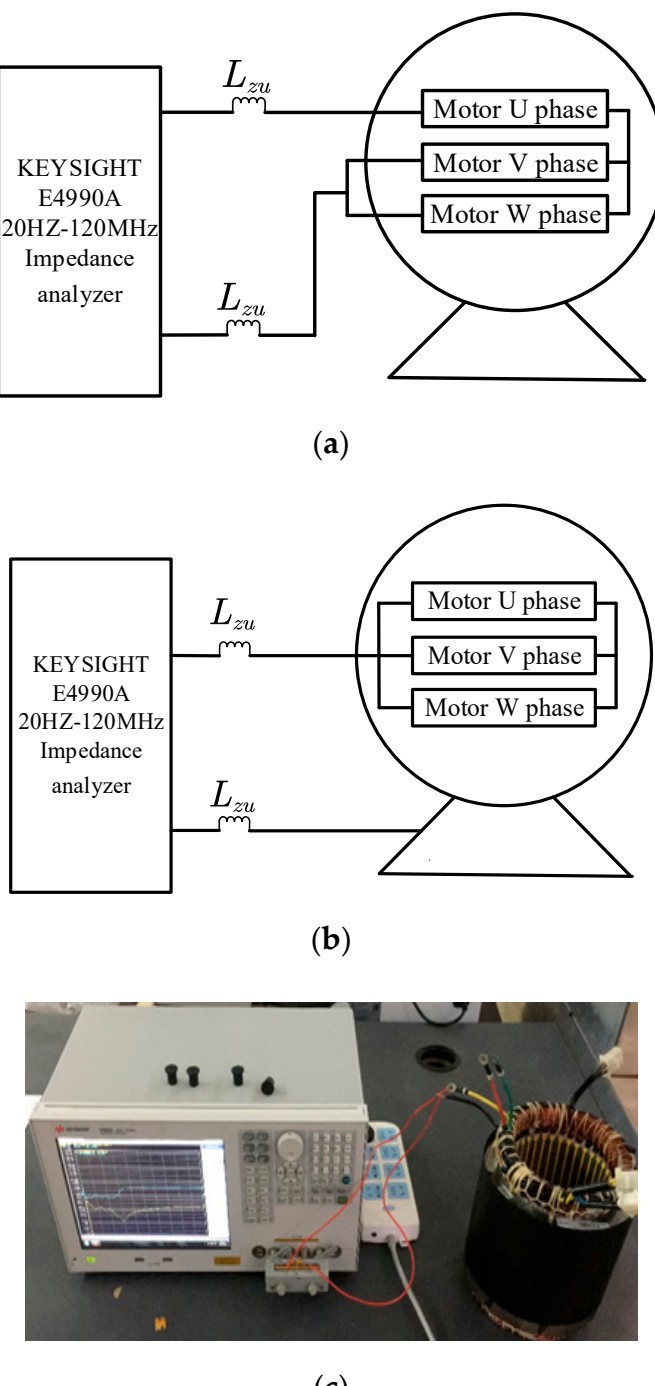

**Figure 5.** Common mode and differential mode impedance measurement wiring diagram of synchronous machine: (**a**) common mode impedance measurement wiring; (**b**) differential mode impedance measurement wiring; and (**c**) measurement site map.

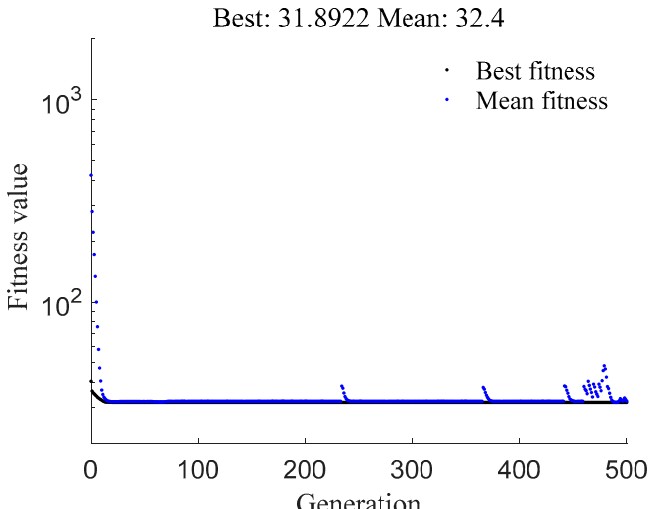

**Figure 6.** Fitness curve of high frequency lumped parameter identification.

Table 1 shows the results of parameter identification. The common mode and differential mode impedance curves of the machine are simulated using the identified circuit parameters and are then compared with the measured common mode and differential mode impedance curves, as shown in Figure 7. The results show that the trend of the identification results is the same as the measured results in the range of 20 Hz–30 MHz, and the curves in the low and medium frequency bands are in good agreement. However, there may be some errors between the identification results and the measured values in the high-frequency band due to the existence of measurement noise and the inherent defects of the single-phase parameter model. However, the model still has specific application scenarios in engineering.

**Table 1.** Parameter identification results.

| Parameter | Value | Parameter | Value | Parameter | Value |
|-----------|-------|-----------|-------|-----------|-------|
| $R_1$ | 166.404 Ω | $C_1$ | 2.1884 pF | $L_d$ | 56.30 mH |
| $L_1$ | 0.01070 H | $R_{Cu}$ | 2.4601 Ω | $R_e$ | 1.0238 kΩ |
| $C_{g2}$ | 2.8184 nF | $R_{g2}$ | 5 Ω | $C_{g1}$ | 95.152 nF |
| $R_{g1}$ | 20.4828 Ω | $L_{zu}$ | 0.2999 μH | | |

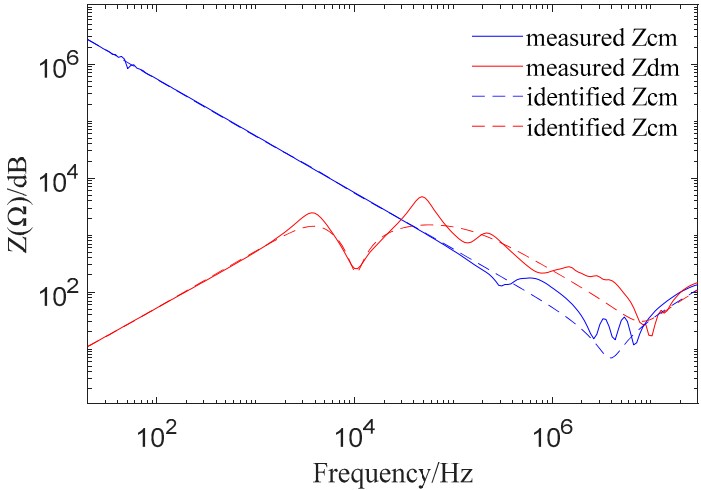

**Figure 7.** Comparison of impedance characteristic curve identified by the optimized algorithm and state space with actual impedance characteristic curve (20 Hz–30 MHz).

## 4. Contrast Experiment

As mentioned above, finite element analysis of a winding structure is complex and it is challenging to achieve the desired effect; multi-unit circuits are not conducive to measuring the common mode and differential mode impedance [12,13], so they are not compared here. Figure 8 shows a comparison of the impedance characteristics. Under the condition that the high frequency lumped parameter model is used as the single-phase winding model of synchronous machine, 2,3 represents the parameters identification based on the proposed method and engineering exploratory calculation method [14,15]. The 1 represents the black box model as a result of the single-phase model parameter identification of synchronous machine winding [9–11].

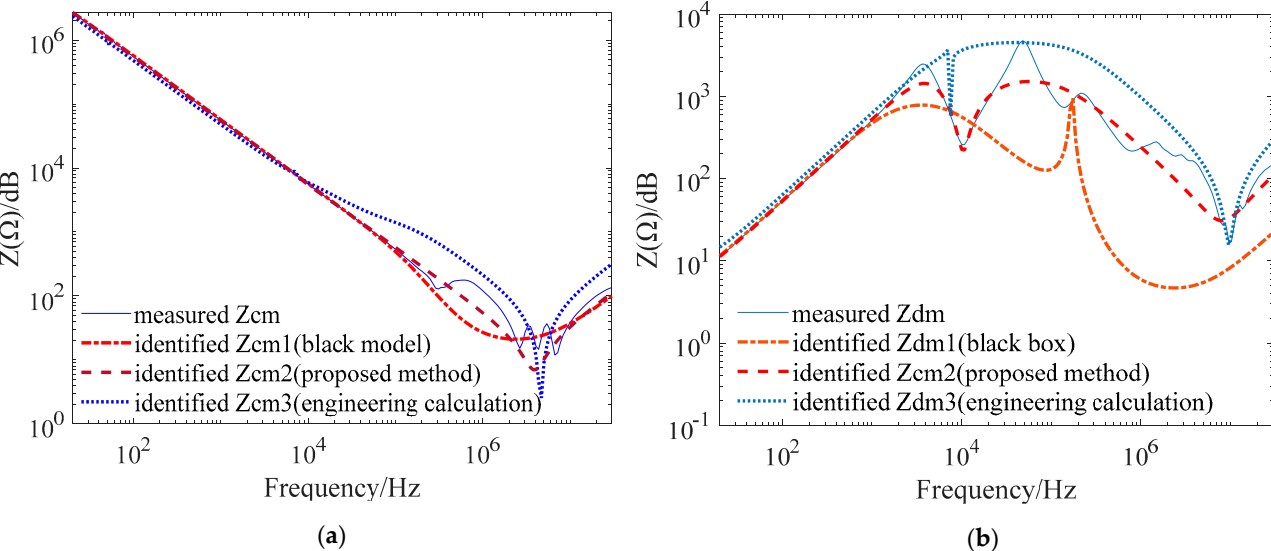

**Figure 8.** Comparison diagram of impedance characteristics: (**a**) comparison of common mode impedance characteristic curves; (**b**) comparison of differential mode impedance characteristic curves.

Mean absolute percentage error (MAPE) can be used to evaluate the quality of the model. Root mean square error (RMSE) is sensitive to the maximum or minimum error in a group of results and can also reflect the identified curve accuracy [27,28]. With the same measured curve, the smaller the MAPE and RMSE are, the more accurate the identified curve. Therefore, this paper uses MAPE and RMSE to evaluate the accuracy of each identified curve, and the expression is as follows:

$$E_{MAPE} = \frac{1}{n} \sum_{i=1}^{N} \frac{|Z_{measure} - Z|}{Z_{measure}} \times 100\% \tag{12}$$

$$E_{RMSE} = \sqrt{\frac{1}{n} \sum_{i=1}^{n} (Z_{measure} - Z)^2} \tag{13}$$

where $n$ is the sampling number of the measurement curve, and $Z_{measure}$ and $Z$ are the measured value and the impedance simulation value under each method, respectively.

Table 2 shows the MAPE and RMSE results under three methods. From Table 2, it can be seen that the proposed method's MAPE and RMSE values calculated between the simulated impedance characteristic curve ($Z_{cm}$ and $Z_{dm}$) and the measured curve are the smallest compared with those of the other two methods. Therefore, it can be said that the proposed method is superior to the other two methods, and its accuracy is the highest.

**Table 2.** The results of MAPE and RMSE of the three methods.

| Parameter | Black Box | Proposed Method | Engineering Calculation Method |
|---|---|---|---|
| $Z_{cm}$(MAPE) | 102.0924 | 21.5944 | 27.4624 |
| $Z_{cm}$(RMSE) | 63,880.07 | 13,758.19 | 21,355.57 |
| $Z_{dm}$(MAPE) | 49.3585 | 41.202 | 65.9571 |
| $Z_{dm}$(RMSE) | 1606.95 | 1484.99 | 2051.712 |

It can be seen from the Figure 8 that compared with the other two methods, the optimization algorithm and the model under the state space have the highest degree of agreement, and can better contain the information of each resonance point of the actual curve. Compared with the engineering exploratory calculation method, the parameter identification results are not subjective because the engineering exploratory calculation is an estimates. Additionally, there are certain calculation flaws when calculating the parameters of circuit elements in the low-frequency band; elements which have less contribution to the low-frequency band are ignored, which is the same as calculating the parameters of circuit elements in the high-frequency band. The use of the optimization algorithm has a certain degree of robustness, and it has strong adaptability to actual measured impedance characteristic curve data. Compared with the black box model, the proposed single-phase high-frequency lumped parameter model has a certain physical meaning, but the black box model only contains the information of each resonance point without any physical meaning. However, in Figure 8, this method also does not effectively include the information of each resonance point, which is due to the measurement of the frequency band is too broad [9–11]. In conclusion, the results in Table 2 and Figure 8 show that the proposed method is superior to the other two methods.

## 5. Conclusions

This paper presents a modeling method of synchronous machine winding based on broadband characteristics. The common mode and differential mode measurement methods are used to measure the port impedance characteristics of a machine. On this basis, the proposed method is used to identify equivalent parameters. Finally, the common mode and differential mode impedance characteristic curve are simulated and compared with the actual measurement curve, and the following conclusions are obtained:

1.  Based on the state space and the optimization algorithm's parameter identification, there is a relatively high degree of agreement between the simulated curve and the measured curve, which indicates that the proposed modeling method is adaptive and robust to the measured data. On the other hand, if it chooses to use the state space model, it can modify the output Equation (9) to adapt to different measured external characteristic curves, so the method has potent portability.
2.  Compared with the black box model, the proposed method has a certain physical meaning to the broadband equivalent circuit model, as such it has better applicability, and the method does not need to master the detailed design parameters of the synchronous machine, it is simple to implement, and meets the engineering accuracy requirements. Compared with traditional engineering exploratory calculation parameters, the use of particle swarm optimization to identify parameters overcomes the subjectivity of selecting feature points in engineering calculations. Furthermore, the use of a simple particle swarm algorithm shows that the parameter identification method has a certain universality, and the completion of parameter identification does not require more complex algorithms.
3.  Although the single-phase lumped parameter circuit model proposed in this paper is greatly affected by noise at a high frequency, it can still be used to research the basis of high-frequency problems of synchronous machines, such as predicting common mode current, electromagnetic interference of drive system, etc.

**Author Contributions:** Y.C. derived formulas, program verification and data analyses, wrote and revised manuscript; X.J. contributed data analysis, literature review, and grammar corrections; Z.Z. provided an experimental platform and ideas. All authors have read and agreed to the published version of the manuscript.

**Funding:** This research was funded by the National Natural Science Foundation of China, grant number 51807166, the Natural Science Foundation of Chongqing, grant number cstc2019jcyj-msxmX0236, and the Venture and Innovation Support Program for Chongqing Overseas Returnees, grant number cx2019123.

**Institutional Review Board Statement:** All procedures performed in studies involving human participants were in accordance with the ethical standards of the institutional.

**Informed Consent Statement:** Informed consent was obtained from all subjects involved in the study.

**Data Availability Statement:** No data available.

**Conflicts of Interest:** The authors declare no conflict of interest.

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
