# Peer review of "Synchronous Machine Winding Modeling Method Based on Broadband Characteristics"

_applsci, doi:10.3390/app11104631_

Round 1
Reviewer 1 Report
The paper is, in my opinion, a good example of the properly prepared journal article with enough effort of the authors both in scintific content as well as editorial items. However I have some remarks to the current form of the article as below:
1) Last sentence in Abstract is 8 lines long. Understanding such long sentence is hardly probable. Please rewrite dividing it into 2-3 shorter sentences. In general, please try to verify the text from the point of view of long sentences. They should be more compact and shorter.
2) In Introduction, presenting the methods for winding modeling the authors should complement them by adding the proper references in each case when they are bulleted.
3) Fig. 1 - Rg is not explained in the text. Please clearly explain also what means subscript 1 and 2 for Rg and Cg.
4) Please revise the fragment in lines 148-156. There are some repetitions and some phrases are too trivial. Please provide some literature examples or more deep explenation of your point of view for chosing PSO algorithm.
5) It is suggested that Figures follow the text where they are introduced. I mean the Figs 2 and 3 (first text - then figures). bThe same concerns Table 1 and Fig. 6.
6) Lines 228-229: What the authors have in mind writting "the inherent defects of the single-phase parameter model"?
Author Response
Responses to Comments from Editor and Reviewers
Dear Editor and Reviewers:
Thank you very much for your comments and thoughtful suggestions on our previous submitted manuscript-1204839 entitled “Synchronous machine winding modeling method based on broadband characteristics”. These comments are all valuable and quite helpful for revising and improving our paper, as well as important for our further research. We have studied these comments carefully and then made major modifications on our manuscript. The point-to-point responses and explanations for all comments are listed in the following while the modified words and sentences are marked with red. The majorly revised contents are marked with red in the revised manuscript. We hope that these revisions can clearly explain your concerns and meet your requirement. And we hope that the revised manuscript is now suitable for publication. If you have any further questions about this paper, please contact us without hesitation.
We are looking forward to hearing from you soon.
Best regards
Zhongyong Zhao
5.6, 2021
Response to Reviewer 1 Comments
The paper is, in my opinion, a good example of the properly prepared journal article with enough effort of the authors both in scintific content as well as editorial items. However I have some remarks to the current form of the article as below:
Point 1: Last sentence in Abstract is 8 lines long. Understanding such long sentence is hardly probable. Please rewrite dividing it into 2-3 shorter sentences. In general, please try to verify the text from the point of view of long sentences. They should be more compact and shorter.
Response 1: Thank you for the suggestion. The sentence has been modified to "The experimental results show that: The proposed modeling method identifies the impedance curve of the circuit parameters with a high degree of agreement with the measured impedance curve, which indicates that the modeling method is feasible. In addition, the comparative experimental results show that: Compared with the engineering exploratory calculation method, the proposed parameter identification method has stronger adaptability to the measured data and certain robustness. Compared with the black box model, the parameters of the proposed model have a certain physical meaning, and the agreement with the actual impedance characteristic curve is higher than that of the black box model.". Please find it in the revised manuscript.(Line 16-22)
Point 2: In Introduction, presenting the methods for winding modeling the authors should complement them by adding the proper references in each case when they are bulleted.
Response 2: Thank you for the suggestion. The sentence has been cited the proper references and modified to "At present, there are three broadly applicable methods for winding modeling:1. Use the resonant unit of the black box model to simulate the information of each resonant point, and then cascade each resonant unit to construct a circuit model [9-11]. 2. The model is established by using the physical properties of the machine, such as magnetic flux leakage, copper consumption, and leakage current, etc [12-14]. 3. The whole machine winding's physical structure is analyzed by the finite element method, and then the winding is modeled by using the analysis results [15]. " Please find it in the revised manuscript. (Line 41-47)
Reference:
- Hong S.; Research on conducted electromagnetic interference of electric vehicle motor drive system. Chongqing University. 2012. (in Chinese)
- Idir N.; Weens Y.; Moreau M.; Franchaud J. High-Frequency Behavior Models of AC Motors. IEEE Trans. on Magn. 2009, 45, 1, 133-138.
- Quandi W.; Hong S.;Yali Z. Rapid modeling method of motor broadband equivalent circuit model . Chongqing: Journal of Chongqing University, 2012(in Chinese)
- Blánquez F.; A. Platero C.; Rebollo E.; Blánquez F. Evaluation of the applicability of FRA for inter-turn fault detection in stator windings. 2013 9th IEEE International Symposium on Diagnostics for Electric Machines, Power Electronics and Drives (SDEMPED), Valencia, Spain, 2013, 177-182.
- Shabestary M.; J. Ghanizadeh A.; B. Gharehpetian G.; Agha-Mirsalim M. Ladder Network Parameters Determination Considering Nondominant Resonances of the Transformer Winding. IEEE Trans. Power Deliver. 2014, 29, 108-11.
- Schinkel M.; Weber S.; Guttowski S.; John W.; Reichl H. Efficient HF modeling and model parameterization of induction machines for time and frequency domain simulations. Twenty-First Annual IEEE Applied Power Electronics Conference and Exposition, TX, USA. 2006.
- Mohammed O.; Ganu S.; Abed N.; Liu S.; Liu Z. High frequency PM synchronous motor model determined by FE analysis. IEEE Trans. on Magn. 2006, 42, 4, 1291-1294.
Point 3: Fig. 1 - Rg is not explained in the text. Please clearly explain also what means subscript 1 and 2 for Rg and Cg.
Response 3: Sorry to have confused you, it was the author's negligence. The sentence has been modified to "according to the reference [17], Rg1 and Cg1 represent the parasitic resistance and capacitance between the stator winding and the motor frame; Rg2 and Cg2 represent the parasitic resistance and capacitance between the stator neutral and motor frame ." Please find it in the revised manuscript. (Line 107-110)
Reference:
- Wang L.; Ngai-Man Ho C.; Canales F.; Jatskevich J. High-Frequency Modeling of the Long-Cable-Fed Induction Motor Drive System Using TLM Approach for Predicting Overvoltage Transients. IEEE Trans Power Electr. 2010, 25, 10, 2653-2664.
Point 4: Please revise the fragment in lines 148-156. There are some repetitions and some phrases are too trivial. Please provide some literature examples or more deep explenation of your point of view for chosing PSO algorithm.
Response 4: Thank you for your suggestion. The sentence has been modified to "When identifying the model's parameters, the theoretical model with the experimental data are usually combined to identify the parameter value [19-26]. As an essential method of parameter identification, the optimization algorithm has a strong search performance for a specific range parameter, and it has high adaptability to practical problems [19,21,23-24]. In addition, it can visualize the performance of identification results through the value of fitness function. Therefore, this paper chooses an optimization algorithm as a tool for parameter identification.
The particle swarm optimization (PSO) algorithm is selected as the identification algorithm. The circuit constructed in this paper is relatively simple, and it does not need an algorithm with strong search ability and complicated calculation. Second, because the PSO algorithm has few adjustable hyperparameters, the algorithm structure is relatively simple compared with other intelligent algorithms (simulated annealing algorithm, genetic algorithm, immune algorithm, etc.). It is also easy to implement in engineering with faster convergence speed [20,22,23]. Third, the algorithm structure of PSO is simple and it has high universality. Fourth, there are a large number of identification parameters in this study. The number of solutions of the problem can be directly taken as the dimension of the particle in PSO algorithm. Thus, compared with other algorithms, PSO algorithm is less affected by the dimension of the problem [20,22,23]."(Line 167-184)
Reference:
- T. Cari E.; F. C. Alberto L.; Erlich I. Assessment of model parameters to identify an equivalent wind power plant. 2015 IEEE Eindhoven PowerTech. 2015, 1-5.
- Zhang H.; Hao J. A method using PSO to optimize and identify error parameter based on local state estimation. 2017 IEEE Power & Energy Society General Meeting. 2017, 1-5.
- Li M.; Ma Y. Parameter Identification of DC Motor based on Compound Least Square Method. 2020 IEEE 5th Information Technology and Mechatronics Engineering Conference (ITOEC). 2020, 1107-1111.
- Peng W.; Yang Z.; Liu C.; Xiu J.; Zhang Z.; An Improved PSO Algorithm for Battery Parameters Identification Optimization Based on Thevenin Battery Model. 2018 5th IEEE International Conference on Cloud Computing and Intelligence Systems (CCIS). 2018, 295-298.
- Maharana H.; K. Dash S. Comparative Optimization Analysis of Ramp Rate Constriction Factor Based PSO and Electro Magnetism Based PSO for Economic Load Dispatch in Electric Power System. 2019 International Conference on Applied Machine Learning (ICAML). 2019, 63-68.
- Tang, H.; Guo, X.; Xie, L.; Xue, S. Experimental Validation of Optimal Parameter and Uncertainty Estimation for Structural Systems Using a Shuffled Complex Evolution Metropolis Algorithm. Sci. 2019, 9, 4959.
- Ariza, H.E.; Correcher, A.; Sánchez, C.; Pérez-Navarro, Á.; García, E. Thermal and Electrical Parameter Identification of a Proton Exchange Membrane Fuel Cell Using Genetic Algorithm. Energies 2018, 11, 2099.
- Bui, H.-B.; Nguyen, H.; Choi, Y.; Bui, X.-N.; Nguyen-Thoi, T.; Zandi, Y. A Novel Artificial Intelligence Technique to Estimate the Gross Calorific Value of Coal Based on Meta-Heuristic and Support Vector Regression Algorithms. Sci. 2019, 9, 4868.
Point 5: It is suggested that Figures follow the text where they are introduced. I mean the Figs 2 and 3 (first text - then figures). The same concerns Table 1 and Fig. 6.
Response 5: Thank you for your suggestion. We have repositioned the corresponding figures and tables. Please find them in the revision manuscript.
Point 6: Lines 228-229: What the authors have in mind writting "the inherent defects of the single-phase parameter model"?
Response 6: Sorry to have confused you. In fact, the proposed model is a relatively simplified equivalent circuit model. It represents each frequency band characteristic of the winding with these basic circuit elements, which is in fact a rough simulation of the single phase of the synchronous machine. Instead of taking into account the mutual inductance relationship between the windings, the proposed model simplifies its effect to the inductance of each single-phase model. Besides, from the high frequency point of view, the skin effect of the motor is also not considered. Even, the author considers that the circuit model parameters of each phase model are actually different because of the different placement positions of winding or the uneven characteristics of the material.
However, the effect of these inherent defects are obvious in the high frequency band and not obvious in the low frequency band, so the impedance characteristic curves of the model in the low frequency band have a high degree of agreement with the measured curve.
If the model is applied to the electromagnetic interference prediction of the machine(1Hz-10Mhz), the very high frequency band(10MHz-30MHz) [11] provided in this paper is not needed. This paper only provides a synchronous machine modeling method which is different from other methods. It can be used to solve the problem that the high frequency motor equivalent model needs to be used in the future. Next, the author will focus the research on overcoming the shortcomings mentioned above.
Reference:
- Quandi W.; Hong S.;Yali Z. Rapid modeling method of motor broadband equivalent circuit model . Chongqing: Journal of Chongqing University, 2012(in Chinese)

Reviewer 2 Report
In this study Authors presented the synchronous machine winding modeling method based on broadband characteristics. The subject of the research is important and actual, however I do not see the novelty of this paper. Also the methodolgy should be improved and presented in more details. The results section should be extended and widely discussed. Below please find my detailed comments:
- the all symbols used in the figures should be explained in the figure captions.
- Lines 110-112: please change the style of these sentences.
- please check and improve English. I suggest to do it with help of native speaker.
- line 151-156: please rephrase this sentence, and break it down into a few shorter sentences. In it's present form it is hard to understand the meaning.
- In the methodology section the scheme presenting the methodology and scope of the research step by step,
- the methodology section should be presented in more details
- please
- please improve the quality of the figures, especially the font should be bigger, because now they are to small and it is hard to read the axes and their titles,
- Figure 7, the curves from different methods should be better explained in the legend or the figure caption should contain explenation which curve on the figure represents which method
- From the presented results I can not see the novelty and the Authors contribution to the state of the art,
- please explain, why your model is better from others, from figure 7 it is not obvious and clear,
- the results section should be extended
- explain how you build the fitness function and what is the meaning of the figure 5
Author Response
Responses to Comments from Editor and Reviewers
Dear Editor and Reviewers:
Thank you very much for your comments and thoughtful suggestions on our previous submitted manuscript-1204839 entitled “Synchronous machine winding modeling method based on broadband characteristics”. These comments are all valuable and quite helpful for revising and improving our paper, as well as important for our further research. We have studied these comments carefully and then made major modifications on our manuscript. The point-to-point responses and explanations for all comments are listed in the following while the modified words and sentences are marked with red. The majorly revised contents are marked with red in the revised manuscript. We hope that these revisions can clearly explain your concerns and meet your requirement. And we hope that the revised manuscript is now suitable for publication. If you have any further questions about this paper, please contact us without hesitation.
We are looking forward to hearing from you soon.
Best regards
Zhongyong Zhao
5.6, 2021
Response to Reviewer 2 Comments
In this study Authors presented the synchronous machine winding modeling method based on broadband characteristics. The subject of the research is important and actual, however I do not see the novelty of this paper. Also the methodolgy should be improved and presented in more details. The results section should be extended and widely discussed. Below please find my detailed comments:
Point 1: the all symbols used in the figures should be explained in the figure captions.
Response 1: Thank you for your suggestions. The author has explained the unexplained symbols in the figures. Please find it in the revised manuscript.
Point 2: Lines 110-112: please change the style of these sentences.
Response 2: Thank you for your suggestions. The sentence has been modified to " In order to obtain the value of each circuit parameters described above, the first step is to find the circuit equations that contain each parameter in the circuit topology shown in Fig.2(b) and (c). When the above circuits are analyzed using the state space model, the capacitance voltage and inductance current in the single-phase lumped parameter circuit model are selected as the state variables. The KCL and KVL equations are written for the whole circuit, and these equations are finally integrated into matrices. "(Line 125-130)
Point 3: please check and improve English. I suggest to do it with help of native speaker.
Response 3: Thank you for your suggestion. We updated the manuscript by delivering our text to a native English speaker, who has revised the manuscript's English to make the work easy to follow.
Point 4: line 151-156: please rephrase this sentence, and break it down into a few shorter sentences. In it's present form it is hard to understand the meaning.
Response 4: Thank you for your suggestion. The sentence has been modified to " When identifying the model's parameters, the theoretical model with the experimental data are usually combined to identify the parameter value [19-26]. As an essential method of parameter identification, the optimization algorithm has a strong search performance for a specific range parameter, and it has high adaptability to practical problems [19,21,23-24]. In addition, it can visualize the performance of identification results through the value of fitness function. Therefore, this paper chooses an optimization algorithm as a tool for parameter identification.
The particle swarm optimization (PSO) algorithm is selected as the identification algorithm. The circuit constructed in this paper is relatively simple, and it does not need an algorithm with strong search ability and complicated calculation. Second, because the PSO algorithm has few adjustable hyperparameters, the algorithm structure is relatively simple compared with other intelligent algorithms (simulated annealing algorithm, genetic algorithm, immune algorithm, etc.). It is also easy to implement in engineering with faster convergence speed [20,22,23]. Third, the algorithm structure of PSO is simple and it has high universality. Fourth, there are a large number of identification parameters in this study. The number of solutions of the problem can be directly taken as the dimension of the particle in PSO algorithm. Thus, compared with other algorithms, PSO algorithm is less affected by the dimension of the problem [20,22,23]." (Line 167-184)
Reference:
- T. Cari E.; F. C. Alberto L.; Erlich I. Assessment of model parameters to identify an equivalent wind power plant. 2015 IEEE Eindhoven PowerTech. 2015, 1-5.
- Zhang H.; Hao J. A method using PSO to optimize and identify error parameter based on local state estimation. 2017 IEEE Power & Energy Society General Meeting. 2017, 1-5.
- Li M.; Ma Y. Parameter Identification of DC Motor based on Compound Least Square Method. 2020 IEEE 5th Information Technology and Mechatronics Engineering Conference (ITOEC). 2020, 1107-1111.
- Peng W.; Yang Z.; Liu C.; Xiu J.; Zhang Z.; An Improved PSO Algorithm for Battery Parameters Identification Optimization Based on Thevenin Battery Model. 2018 5th IEEE International Conference on Cloud Computing and Intelligence Systems (CCIS). 2018, 295-298.
- Maharana H.; K. Dash S. Comparative Optimization Analysis of Ramp Rate Constriction Factor Based PSO and Electro Magnetism Based PSO for Economic Load Dispatch in Electric Power System. 2019 International Conference on Applied Machine Learning (ICAML). 2019, 63-68.
- Tang, H.; Guo, X.; Xie, L.; Xue, S. Experimental Validation of Optimal Parameter and Uncertainty Estimation for Structural Systems Using a Shuffled Complex Evolution Metropolis Algorithm. Sci. 2019, 9, 4959.
- Ariza, H.E.; Correcher, A.; Sánchez, C.; Pérez-Navarro, Á.; García, E. Thermal and Electrical Parameter Identification of a Proton Exchange Membrane Fuel Cell Using Genetic Algorithm. Energies 2018, 11, 2099.
- Bui, H.-B.; Nguyen, H.; Choi, Y.; Bui, X.-N.; Nguyen-Thoi, T.; Zandi, Y. A Novel Artificial Intelligence Technique to Estimate the Gross Calorific Value of Coal Based on Meta-Heuristic and Support Vector Regression Algorithms. Sci. 2019, 9, 4868.
Point 5: In the methodology section the scheme presenting the methodology and scope of the research step by step, the methodology section should be presented in more details
Response 5: Thank you for your suggestion. In order to make the method more clear and detailed, the author adds a flow chart of proposed method in the methodology part of the manuscript.(Line 99-102)
Figure 1. Flow chart of proposed method
Point 6: please improve the quality of the figures, especially the font should be bigger, because now they are to small and it is hard to read the axes and their titles,
Response 6: Thank you for your suggestion. All images have been resized. The font has been adjusted. Please find it in the revised manuscript.
Point 7: Figure 7, the curves from different methods should be better explained in the legend or the figure caption should contain explenation which curve on the figure represents which method.
Response 7: Thank you for your suggestion. The author has added the legends to Fig. 8.
(a)
(b)
Figure 8. Comparison diagram of impedance characteristics: (a) Comparison of common mode impedance characteristic curves; (b) Comparison of differential mode impedance characteristic curves.
Point 8: From the presented results I can not see the novelty and the Authors contribution to the state of the art, please explain, why your model is better from others, from figure 7 it is not obvious and clear, the results section should be extended
Response 8: Thank you for your suggestion. This paper provides a simple and fast synchronous machine modeling method which is different from other methods. It uses heuristic algorithm to identify and search the circuit parameters without a need to know the specific structural parameters of machine. The parameters in the model also have a certain practical physical significance to reflect the actual situation of the internal winding without losing a certain degree of accuracy.
Compared with the traditional common mode differential mode model [12-13], this paper widens the frequency range of its characteristic curve, and provides a new parameter identification method. The parameter identification method is more robust to the experimental data, and does not have any subjectivity, so that the simulated impedance characteristic curve is closer to the actual measurement curve. Compared with the black box model [14-15], the model presented in this paper has more practical significance and higher consistency for the measured impedance characteristic curve.
In addition, the author has added new indicators to judge the quality and accuracy of the three methods, so that the results are more direct.
Mean absolute percentage error (MAPE) can be used to evaluate the quality of the model. Root mean square error (RMSE) is sensitive to the maximum or minimum error in a group of results and can also reflect the identified curve accuracy [27,28]. With the same measured curve, the smaller the MAPE and RMSE, the more accurate the identified curve. Therefore, this paper uses MAPE and RMSE to evaluate the accuracy of each identified curve, and the expression is as follows:
(12)
(13)
where n is the sampling number of the measurement curve, Zmeasure and Z are the measured value and the impedance simulation value under each method respectively.
Tab.2 is the MAPE and RMSE results under three methods. From Tab.2, it can be seen that the proposed method’s MAPE and RMSE values calculated between the simulated impedance characteristic curve (Zcm and Zdm) and the measured curve are the smallest compared with those of other two methods. Therefore, it can be said that the proposed method is superior to the other two methods, and its accuracy is the highest.(Line 286-303)
Table 2. The results of MAPE and RMSE of the three methods
|
parameter |
Black box |
Proposed method |
Engineering calculation method |
|
Zcm(MAPE) |
102.0924 |
21.5944 |
27.4624 |
|
Zcm(RMSE) |
63880.07 |
13758.19 |
21355.57 |
|
Zdm(MAPE) |
49.3585 |
41.202 |
65.9571 |
|
Zdm(RMSE) |
1606.95 |
1484.99 |
2051.712 |
Reference:
- Blánquez F.; A. Platero C.; Rebollo E.; Blánquez F. Evaluation of the applicability of FRA for inter-turn fault detection in stator windings. 2013 9th IEEE International Symposium on Diagnostics for Electric Machines, Power Electronics and Drives (SDEMPED), Valencia, Spain, 2013, 177-182.
- Shabestary M.; J. Ghanizadeh A.; B. Gharehpetian G.; Agha-Mirsalim M. Ladder Network Parameters Determination Considering Nondominant Resonances of the Transformer Winding. IEEE Trans. Power Deliver. 2014, 29, 108-11.
- Schinkel M.; Weber S.; Guttowski S.; John W.; Reichl H. Efficient HF modeling and model parameterization of induction machines for time and frequency domain simulations. Twenty-First Annual IEEE Applied Power Electronics Conference and Exposition, TX, USA. 2006.
- Mohammed O.; Ganu S.; Abed N.; Liu S.; Liu Z. High frequency PM synchronous motor model determined by FE analysis. IEEE Trans. on Magn. 2006, 42, 4, 1291-1294.
- Zengping W.; Bing Z.; Weijia J.; Xin G.; Xiaobing L. Short term load forecasting method based on gru-nn model . Power system automation. 2019,43, 53-58. (in Chinese)
- Sun W.; Zhang Xing. Application of self-organizing combination forecasting method in power load forecast. 2007 International Conference on Wavelet Analysis and Pattern Recognition, 2007.
Point 9: explain how you build the fitness function and what is the meaning of the figure 5
Response 9: Thank you for your suggestion.
The impedance is sampled when the frequency is equal to the frequency of each point of the actual measurement curve. Then a least-square optimization problem is formed between the impedance and the actual measurement value. The least-square optimization problem is taken as a fitness function. The smaller the value of the fitness function is, the closer the simulation curve is to the actual value. For the constructed model, a fitness function is constructed between the result of Equation and the measurement curve:
(11)
where f is the frequency, Zcm, Zcm1, Zdm, and Zdm1 are the common mode impedance measurement value and identification value, and the differential mode impedance measurement value and identification value, respectively. (Line 166-201)
Sorry to confuse you. The author did not elaborate on the meaning of Fig. 6.
The change in the fitness function is shown in Fig. 6. Fig. 6 shows the fitness Equation (10) trend between the impedance characteristic curve formed by the optimal solution corresponding to each iteration and the actual measured value during the iteration process. The smaller the fitness function value, the closer the impedance characteristic curve simulated by the optimal solution is to the measured curve [19]. According to Fig. 6, it can be seen that in the initial stage of PSO search, the fitness of PSO is sharply reduced, which indicates that the impedance value calculated by the parameters searched by the algorithm at each frequency point is rapidly close to the value of the measured curve at each frequency point. However, in the later stage of the algorithm, there is no change in the fitness, which indicates that the local change of the parameters identified in the later stage of the algorithm under the previous search results cannot significantly affect the change of the fitness value. This phenomenon also indicates that the search results of the algorithm are close to the optimal solution. To a certain extent, it shows that PSO is indeed suitable for multi-parameters identification problems and can get the optimal solution close to the actual measurement curve in a small number of iterations. (Line 233-248)
Figure 6. Fitness curve of high frequency lumped parameter identification.
Reference:
- T. Cari E.; F. C. Alberto L.; Erlich I. Assessment of model parameters to identify an equivalent wind power plant. 2015 IEEE Eindhoven PowerTech. 2015, 1-5.

Round 2
Reviewer 1 Report
As I mentioned earlier I like this paper and after corrections made by authors it may be accepted.
I believe that during final processing paper will be slightly improved from the viewpoint of style of English.
Reviewer 2 Report
Dear Authors,
thank you very much for provided changes. All my remarks was addressed. I do not have any further comments.